# Provably Fast Density-Based Clustering in High Dimensions

## Abstract

DBSCAN is a celebrated algorithm for density-based clustering, but its quadratic runtime hinders scalability to large datasets. In recent years, there has been considerable interest in accelerating DBSCAN. However, existing methods either impose additional structure on the data (e.g., low-dimensionality), or lack rigorous runtime and approximation guarantees. Building on a recent work of Okkels et al. (2025), we propose an LSH-based algorithm that achieves the first *provably* subquadratic runtime for approximate DBSCAN on arbitrary high-dimensional datasets. Empirically, our algorithm delivers a significant speedup over the standard DBSCAN on a variety of benchmarks while incurring only small error. We also show that our approach naturally yields a subquadratic-time approximation of HDBSCAN (a popular hierarchical variant). Complementing our algorithms, we prove quadratic-time lower bounds for exact DBSCAN and HDBSCAN, showing that subquadratic runtimes are only possible with approximation.

## 1 Introduction

This paper is on *density-based clustering*, a core problem in data science and machine learning. While center-based clustering methods (e.g., optimizing the $k$-means objective) group points by their proximity to a fixed set of centers, density-based clustering seeks to identify connected regions of high density—potentially with complicated, non-convex shapes—separated by low-density areas. Due to this flexibility, density-based clustering has found applications across a broad range of fields, from genomic data analysis (Edla & Jana, 2012) to object detection in autonomous driving (Wagner et al., 2015).

The most celebrated algorithm for density-based clustering is DBSCAN (Ester et al., 1996). Given a dataset of $n$ points in a metric space, DBSCAN proceeds in two main phases. First, it identifies a set of *core points*, those whose $\varepsilon$-neighborhoods (for some fixed $\varepsilon > 0$) contain a specified number of dataset points (this threshold is denoted MinPts[1]). Second, it constructs a graph on the set of core points, where an edge is drawn between any two points that are within distance $\varepsilon$. The connected components of this graph are returned as the clusters, while all non-core points are marked as *noise*. A straightforward implementation of DBSCAN requires $\Theta(n^2)$ distance computations in each phase, which is often impractical given the size of modern datasets. This has led to considerable interest over the past decade in designing algorithms which simulate DBSCAN in *subquadratic* time, particularly for datasets in the Euclidean space $\mathbb{R}^d$.

Gunawan (2013) showed that in the Euclidean plane ($d = 2$), DBSCAN can be simulated in $O(n \log n)$ time. Interestingly, this near-linear runtime does not extend to any higher dimension. Indeed, Gan & Tao (2017) proved that, assuming a widely held conjecture in computational geometry, any algorithm that simulates DBSCAN in $\mathbb{R}^d$ for $d \geq 3$ requires at least $\Omega(n^{4/3})$ time. They complemented this with a randomized algorithm achieving a runtime of $O(n^{2-\frac{2}{\lceil d/2 \rceil + 1} + \delta})$ for arbitrarily small constants $\delta > 0$, and in particular, a runtime of $\tilde{O}(n^{4/3})$ for the case of $d = 3$.

To circumvent the polynomial runtime in $\mathbb{R}^d$, Gan & Tao (2017) also introduced a notion of *approximating* the output of DBSCAN. Informally, a $c$-approximation of DBSCAN (with parameters $\varepsilon$ and MinPts) is any clustering that is coarser than the DBSCAN clustering with parameter $\varepsilon$ but finer

---

[1]The parameter MinPts is generally assumed to be a constant.

than the DBSCAN clustering with parameter $c\varepsilon$. They showed that for any constant dimension and constant approximation factor $c$, one can produce a $c$-approximate clustering in expected time $O(n)$. However, the runtime of their algorithm is exponential in $d$, leaving open the following question:

*Is there an efficient algorithm for $c$-approximate* DBSCAN *in high dimensions?*

This question was recently studied by Okkels et al. (2025). They propose an approach based on *locality-sensitive hashing* (Indyk & Motwani, 1998; Andoni & Razenshteyn, 2015), which gives a $c$-approximation of DBSCAN in time $\tilde{O}(dn^{1+1/(2c^2-1)})$. However, their runtime analysis relies on a critical assumption about the dataset: that the number of points at distance $c\varepsilon$ from any given point is on the order of MinPts. Since it is common to set MinPts $\ll n$, this assumption breaks down in settings of densely-packed clusters. The main contribution of our paper is an LSH-based algorithm that computes a $c$-approximate DBSCAN clustering in subquadratic time without requiring any assumptions on the dataset.

A primary drawback of DBSCAN is that its density parameters are shared across all clusters. This motivated HDBSCAN (Campello et al., 2013), a quadratic-time algorithm for computing hierarchical density-based clusterings. Okkels et al. (2025) posed the question of whether an LSH-based approach could yield an efficient approximation of HDBSCAN. We resolve this question in the affirmative by giving a simple reduction from approximate HDBSCAN to approximate DBSCAN.

**Our Contributions.** We introduce LSH-DBSCAN and LSH-HDBSCAN: faster algorithms for high-dimensional $c$-approximate DBSCAN and HDBSCAN. We prove that our algorithms satisfy the formal approximation guarantee and have a subquadratic runtime on all inputs.

The core point identification step of LSH-DBSCAN closely follows that of Okkels et al. (2025). We first construct an LSH family for the input dataset and hash all the points. To determine if a point is core, we compute its distance to all points in the same bucket. For the cluster formation step, we perform an LSH-assisted breadth-first search where we only look for neighbors of a point within its bucket. The main theoretical result of our paper is the following guarantee.

**Theorem 1.1** (informal; see Theorem 3.1). *Given a set of $n$ points in $\mathbb{R}^d$,* LSH-DBSCAN *returns a $c$-approximate* DBSCAN *clustering with high probability and runs in time $\tilde{O}(dn^{1+1/(2c^2-1)+o(1)})$.*

Our LSH-HDBSCAN algorithm works by making logarithmically many calls to LSH-DBSCAN with decreasing values of $\varepsilon$. It performs an intersection operation at each step to construct the cluster hierarchy. We prove the following guarantee.

**Theorem 1.2** (informal; see Theorem 3.2). *Given a set of $n$ points in $\mathbb{R}^d$,* LSH-HDBSCAN *returns a $c$-approximate* HDBSCAN *hierarchy with high probability and runs in time $\tilde{O}(dn^{1+1/(2c^2-1)+o(1)})$.*

In addition to our algorithms, we provide a reduction from the bichromatic closest pair problem to approximate DBSCAN which gives the following lower bound assuming SETH (a widely held conjecture in complexity theory). For any $\alpha > 0$, there is a $\gamma > 0$ such that computing $(1+\gamma)$-approximate DBSCAN takes $\Omega(n^{2-\alpha})$ time. This means any strongly subquadratic algorithm for DBSCAN must produce an approximation.

We empirically analyze the performance of LSH-DBSCAN with several approximation factors $c$ on four benchmarks: MNIST (Deng, 2012), Fashion-MNIST (Xiao et al., 2017), ALOI (Geusebroek et al., 2005), and GloVe (Pennington et al., 2014). For each benchmark, we measure computation speedup and clustering accuracy relative to exact DBSCAN. To eliminate dependence on machine and implementation details, we measure computation speedup by the number of heavy operations (i.e., hash and distance computations). For completeness, we also include raw runtime speedup results in the appendix. We quantify clustering accuracy by the fraction of misclassified points relative to exact DBSCAN. Predictably, both the speedup and misalignment increase with the approximation factor $c$. Across all benchmarks, LSH-DBSCAN achieves at least a $10\times$ computation speedup with misalignment less than 0.1.

Due to the quadratic runtimes of previous high-dimensional DBSCAN algorithms, it is common practice to perform dimension reduction on the dataset and run DBSCAN on the resulting low-dimensional instance. In the appendix, we include a discussion of this approach with an instructive

example of when it fails, highlighting the importance of faster algorithms for DBSCAN in high dimensions.

**Related Work.** Beyond Okkels et al. (2025), there has been prior work on using LSH to develop practical approaches for DBSCAN (Wu et al., 2007; Shiqiu & Qingsheng, 2019), though these methods lack theoretical guarantees on approximation quality. Sampling-based methods (Jang & Jiang, 2019; Jiang et al., 2020) achieve strong empirical performance. However, their accuracy guarantees hold only for low-dimensional data and rely on strong assumptions about the underlying cluster structure. Other theoretical work on DBSCAN has focused on improvements in low-dimensional settings, particularly through the use of parallelization (Wang et al., 2020).

**Organization of the Paper.** Section 2 contains preliminaries that are used throughout the paper. In Section 3, we present our algorithms LSH-DBSCAN and LSH-HDBSCAN and state our main theoretical results (proofs deferred to Appendix A). Section 4 contains our main experimental results and discussion (additional details in Appendix B).

## 2 PRELIMINARIES

Throughout this paper, we will let $P$ denote a dataset of size $n$ with a metric d. We will use $\Delta$ to denote the aspect ratio of $P$ (the ratio between the the maximum and minimum distance). We will also use the shorthand $m$ to denote the $\mathrm{MinPts}$ parameter of DBSCAN and HDBSCAN.

We use the term *clustering* to refer to a partition of $P$ where some points may be designated as *noise*, i.e. a collection of disjoint clusters together with a (possibly empty) set of noise points.

### 2.1 PRELIMINARIES ON DBSCAN

We start by defining the DBSCAN algorithm (Ester et al., 1996).

**Definition 2.1** (DBSCAN). Given a neighborhood radius $\varepsilon > 0$ and size threshold $m \in \mathbb{N}$, the $(\varepsilon, m)$-DBSCAN *clustering* of $P$ is defined as the output of DBSCAN, the following deterministic algorithm:

1. **Core point identification.** For each $p \in P$, define its $\varepsilon$-neighborhood as $N_\varepsilon(p)$, the set of points $p'$ for which $\mathrm{d}(p', p) \leq \varepsilon$. A point $p \in P$ is called a *core point* if $|N_\varepsilon(p)| \geq m$.

    Let $\mathrm{Core}(P) \subseteq P$ denote the set of all core points. All non-core points are labeled as *noise*.

2. **Cluster formation.** Construct $G_\varepsilon = (\mathrm{Core}(P), E)$, where $(p, p') \in E$ if and only if $p' \in N_\varepsilon(p)$. Return the clustering given by the connected components of $G_\varepsilon$.

The original DBSCAN algorithm also identifies border points: non-core points within distance $\varepsilon$ of a core point. Definition 2.1, sometimes referred to as DBSCAN* (Campello et al., 2013), omits this step for simplicity. All results in this paper easily extend to the original DBSCAN definition.

Following Gan & Tao (2017), we define a notion of *approximating* the output of DBSCAN.

**Definition 2.2** (Refinement). Let $S$ be a set and let $\mathcal{C}, \mathcal{C}'$ be two clusterings of $S$. We say that $\mathcal{C}$ is a *refinement* of $\mathcal{C}'$, denoted $\mathcal{C} \preceq \mathcal{C}'$, if every cluster in $\mathcal{C}$ is a subset of a cluster in $\mathcal{C}'$.

**Definition 2.3** (Approximate DBSCAN, Gan & Tao (2017)). Let $\varepsilon > 0$ and $m \in \mathbb{N}$ be parameters. Given an approximation factor $c \geq 1$, the *c-approximate* $(\varepsilon, m)$-DBSCAN problem asks for a clustering $\mathcal{C}$ of $P$ satisfying the following properties:

- $\mathcal{C}$ is a refinement of the $(\varepsilon, m)$-DBSCAN clustering of $P$.

- The $(\varepsilon/c, m)$-DBSCAN clustering of $P$ is a refinement of $\mathcal{C}$.

This notion of approximation is informally referred to as a "sandwiching" condition: the output clustering $\mathcal{C}$ lies between coarser and finer DBSCAN clusterings. We note that the original definition

of Gan & Tao (2017) sandwiches the clustering in the interval $[\varepsilon, c\varepsilon]$ as opposed to $[\varepsilon/c, \varepsilon]$. These are equivalent under a change of variables. We use our definition since the output of our approximation algorithm is intuitively (and empirically) closer to the coarser clustering.

## 2.2 PRELIMINARIES ON HDBSCAN

We now define the HDBSCAN algorithm (Campello et al., 2013), which is a hierarchical extension of DBSCAN. We first define a notion of cluster hierarchy, which is the output format of HDBSCAN.

**Definition 2.4.** A *cluster hierarchy* on a set $S$ is a sequence $\mathcal{T} = (\mathcal{C}_1, \ldots, \mathcal{C}_L)$, where each $\mathcal{C}_i$ is a clustering of $S$, and

$$\mathcal{C}_1 \preceq \cdots \preceq \mathcal{C}_L.$$

**Definition 2.5** (HDBSCAN). Let $P$ be a dataset with a metric d. Given a parameter $m \in \mathbb{N}$, the $m$-HDBSCAN *hierarchy* on $P$ is defined to be the output of HDBSCAN, the following deterministic algorithm:

1. **Core radius computation.** For $p \in P$, define $\varepsilon(p) := \inf\{\varepsilon \in \mathbb{R} : |N_\varepsilon(p)| \geq m\}$.

2. **Mutual reachability.** For $p, p' \in P$, define $\mathsf{d}_{\mathrm{mr}}(p, p') := \max\{\mathsf{d}(p, p'), \varepsilon(p), \varepsilon(p')\}$

3. **Tree formation.** Compute an MST of the graph on $P$ with edge weights given by $\mathsf{d}_{\mathrm{mr}}$.

4. **Cluster hierarchy formation.** Let $\varepsilon_1 < \cdots < \varepsilon_L$ be the sorted list of all pairwise distances $\{\mathsf{d}(p, p') : p, p' \in P\}$. For each $i \in [L]$, compute the subgraph of the MST consisting of edges with weight at most $\varepsilon_i$, and define $\mathcal{C}_i$ as the clustering of $P$ given by the connected components of this subgraph. Mark any point $p$ with $\varepsilon(p) > \varepsilon_i$ as noise. Return the sequence $\{(\mathcal{C}_i, \varepsilon_i) : i \in [L]\}$.

**Remark 2.1** (Proposition 1 of Campello et al. (2013)). In the $m$-HDBSCAN hierarchy on $P$, the sequence $(\mathcal{C}_1, \ldots, \mathcal{C}_L)$ forms a cluster hierarchy, where $\mathcal{C}_i$ is the $(\varepsilon_i, m)$-DBSCAN clustering of $P$.

Remark 2.1 motivates the following definition of approximate HDBSCAN (de Berg et al., 2017).

**Definition 2.6** (Approximate HDBSCAN). For $m \in \mathbb{N}$ and $c \geq 1$, a cluster hierarchy $\mathcal{T} = (\mathcal{C}_1, \ldots, \mathcal{C}_L)$ on $P$ is a *$c$-approximate $m$-HDBSCAN hierarchy* if there exists a mapping

$$\sigma : (0, \infty) \to [L]$$

such that, for every $\varepsilon > 0$, $\mathcal{C}_{\sigma(\varepsilon)}$ is a $c$-approximate $(\varepsilon, m)$-DBSCAN clustering of $P$.

## 2.3 PRELIMINARIES ON LOCALITY-SENSITIVE HASHING

Here, we define the notion of locality-sensitive hashing, or LSH.

**Definition 2.7** (LSH family, Indyk & Motwani (1998)). Let $(X, \mathsf{d})$ be a metric space, and let $r > 0$, $c \geq 1$, and $p_1, p_2 \in [0, 1]$ be constants. An *$(r, cr, p_1, p_2)$-sensitive hash family* $\mathcal{H}$ is a distribution over hash functions $\boldsymbol{h} : X \to \mathcal{R}$ such that for any $p, q \in X$, one has:

- $\mathsf{d}(p, q) \leq r \implies \Pr_{\boldsymbol{h} \sim \mathcal{H}}[\boldsymbol{h}(p) = \boldsymbol{h}(q)] \geq p_1$.

- $\mathsf{d}(p, q) \geq cr \implies \Pr_{\boldsymbol{h} \sim \mathcal{H}}[\boldsymbol{h}(p) = \boldsymbol{h}(q)] \leq p_2$.

For an integer $t \in \mathbb{N}$, we use $\mathcal{H}^t$ to denote the concatenation of $t$ independent functions from $\mathcal{H}$.

**Definition 2.8** (LSH-friendly metric space). We say that a metric space is *LSH-friendly with quality $\rho$* if there exist parameters $p_1, p_2 \in [0, 1]$ such that, for every $r > 0$ and $c \geq 1$, the space admits an $(r, cr, p_1, p_2)$-sensitive hash family whose quality parameter is $\rho(c) = \log(1/p_1)/\log(1/p_2)$.

**Fact 2.1** (Andoni & Razenshteyn (2015)). *The Euclidean space $\mathbb{R}^d$ is LSH-friendly with quality parameter $\rho(c) = 1/(2c^2 - 1) + o(1)$.*

## 3 FAST ALGORITHMS FOR DBSCAN AND HDBSCAN

In this section, we present subquadratic-time algorithms for approximate DBSCAN and HDBSCAN on arbitrary high-dimensional datasets.

Given a dataset $P$, let $T_{\mathrm{dist}}$ denote the time to compute a distance between two points in $P$. For a given LSH family on $P$, let $T_{\mathrm{hash}}$ denote the time to compute a hash value.

**Theorem 3.1.** *Let $P$ be an $n$-point dataset that is LSH-friendly with quality $\rho$. For any parameters $m \leq n$, $\varepsilon > 0$, $c \geq 1$, and $\delta > 0$, there is an algorithm that runs in expected time*

$$O\left((T_{\mathrm{dist}} + T_{\mathrm{hash}}) \cdot n^{1+\rho} m^{1-\rho} \log(n) \log(n/\delta)\right)$$

*and outputs a $c$-approximate $(\varepsilon, m)$-DBSCAN clustering of $P$, with probability at least $1 - \delta$.*

In particular, for Euclidean datasets $P \subset \mathbb{R}^d$ and constant values[2] of $m$, Theorem 3.1 yields an algorithm that computes a $c$-approximate DBSCAN clustering in time $\tilde{O}(dn^{1+1/(2c^2-1)+o(1)})$.

The hierarchical variant (LSH-HDBSCAN) works by making a logarithmic number of calls to LSH-DBSCAN and therefore inherits the same runtime guarantees up to a logarithmic factor.

**Theorem 3.2.** *Let $P$ be an $n$-point dataset that is LSH-friendly with quality $\rho$ and has aspect ratio $\Delta$. For any parameters $m \leq n$, $c \geq 1$, and $\delta, \gamma > 0$, there is an algorithm that runs in expected time*

$$O\left((T_{\mathrm{dist}} + T_{\mathrm{hash}}) \cdot n^{1+\rho} m^{1-\rho} \log(n) \log(n/\delta) \cdot \log_{1+\gamma} \Delta\right)$$

*and outputs a $c(1 + \gamma)$-approximate $m$-HDBSCAN hierarchy on $P$, with probability at least $1 - \delta$.*

As above, for Euclidean datasets $P \subset \mathbb{R}^d$ and constant values of $m, c, \gamma$, Theorem 3.2 yields an algorithm that computes a $c(1 + \gamma)$-approximate HDBSCAN hierarchy in time $\tilde{O}(dn^{1+1/(2c^2-1)+o(1)})$.

In the following subsections, we present our algorithms LSH-DBSCAN and LSH-HDBSCAN. The proofs of Theorems 3.1 and 3.2 are deferred to Appendix A.

### 3.1 THE LSH-DBSCAN ALGORITHM

We present an algorithm for approximating DBSCAN via locality-sensitive hashing.

**Algorithm overview.** LSH-DBSCAN (Algorithm 1) follows the two-phase structure of DBSCAN.

1. **Core point identification.** As in Okkels et al. (2025), we use an LSH family to compute a subset of all core points at radius $\varepsilon$ that is guaranteed to contain all core points at radius $\varepsilon/c$. This is handled by Algorithm 2.
2. **Cluster formation.** We perform an LSH-assisted breadth-first-search on the $\varepsilon$-neighborhood graph, which ensures that core points within distance $\varepsilon/c$ are placed in the same connected component. This is handled by Algorithm 3.

We assume the existence of an $(\varepsilon/c, \varepsilon, p_1, p_2)$-sensitive hash family $\mathcal{H}$ on $P$ for constants $p_1, p_2$ and quality $\rho = \log(1/p_1)/\log(1/p_2)$. For each hash function $h \in \mathcal{H}$ and point $p \in P$, we define the hash bucket $B_h(p) := \{p' \in P : h(p') = h(p)\}$.

The analysis of LSH-DBSCAN (proof of Theorem 3.1) is deferred to Appendix A.1.

---

**Algorithm 1** LSH-DBSCAN

---

    **Input:** Points $P$; density parameters $(\varepsilon, m)$; approximation factor $c$; failure probability $\delta$
    **Output:** A clustering $C_1, \ldots, C_k$ of $P$
  1: $\widehat{\mathrm{Core}} \leftarrow \textsc{CorePointIdentification}(P, \varepsilon, m, c, \delta)$
  2: Return $\textsc{ClusterFormation}(\widehat{\mathrm{Core}}, \varepsilon, c, \delta)$

---

[2]Prior works on fast DBSCAN (Gan & Tao, 2017; de Berg et al., 2017) assume that $m$ is a constant.

---

**Algorithm 2** CorePointIdentification$(P, \varepsilon, m, c, \delta)$

---

**Input:** Points $P$; density parameters $(\varepsilon, m)$; approximation factor $c$; failure probability $\delta$

**Output:** Set of (approximate) core points $\widehat{\text{Core}}$

1: $\widehat{\text{Core}} \leftarrow \{\}$
2: Compute parameters $K \leftarrow \frac{\log(n/m)}{\log(1/p_2)}$, $T \leftarrow p_1^{-K} \log(2nm/\delta)$
3: Sample $T$ independent hash functions $\mathbf{h}_1, \ldots, \mathbf{h}_T$ from $\mathcal{H}^K$
4: Build all hash tables $B_{\mathbf{h}_1}, \ldots B_{\mathbf{h}_T}$
5: **for** each point $p \in P$ **do**
6:     Check if there are $\geq m$ points in $B_{\mathbf{h}_1}(p) \cup \cdots \cup B_{\mathbf{h}_T}(p)$ that are within distance $\varepsilon$ from $p$
7:     If so, update $\widehat{\text{Core}} \leftarrow \widehat{\text{Core}} \cup \{p\}$
8: **end for**
9: Return the set $\widehat{\text{Core}}$

---

**Algorithm 3** ClusterFormation$(\widehat{\text{Core}}, \varepsilon, c, \delta)$

---

**Input:** Core points $\widehat{\text{Core}}$; density parameter $\varepsilon$; approximation factor $c$; failure probability $\delta$

**Output:** A clustering $C_1, \ldots, C_k$ of $P$

1: Compute parameters $K \leftarrow \frac{\log(|\widehat{\text{Core}}|)}{\log(1/p_2)}$, $T \leftarrow p_1^{-K} \log(2n/\delta)$
2: Sample $T$ independent hash functions $\mathbf{h}_1, \ldots, \mathbf{h}_T$ from $\mathcal{H}^K$
3: Build all hash tables $B_{\mathbf{h}_1}, \ldots B_{\mathbf{h}_T}$
4: Initialize $\ell \leftarrow 1$
5: **while** $\exists p \in \widehat{\text{Core}} \setminus (C_1 \cup \cdots \cup C_{\ell-1})$ **do**
6:     Initialize a queue $Q \leftarrow \{p\}$
7:     Remove $p$ from each table $B_{\mathbf{h}_1}, \ldots, B_{\mathbf{h}_T}$
8:     Initialize a new cluster $C_\ell \leftarrow \{\}$
9:     **while** $|Q| > 0$ **do**
10:         Dequeue a point $u$ from $Q$
11:         Update $C_\ell \leftarrow C_\ell \cup \{u\}$
12:         **for** each point $v$ in $B_{\mathbf{h}_1}(u) \cup \cdots \cup B_{\mathbf{h}_T}(u)$ **do**
13:             If $\text{dist}(u, v) \leq \varepsilon$, add $v$ to $Q$
14:             Remove $v$ from each table $B_{\mathbf{h}_1} \cup \cdots \cup B_{\mathbf{h}_T}$
15:         **end for**
16:     **end while**
17:     Increment $\ell \leftarrow \ell + 1$
18: **end while**
19: Return the clustering $C_1, \ldots, C_k$ where $k := \ell - 1$

---

### 3.2 The LSH-HDBSCAN Algorithm

Here, we present an algorithm for computing an approximation to the output of HDBSCAN.

**Algorithm overview.** Our algorithm LSH-HDBSCAN (Algorithm 4) works by making a logarithmic number of calls to LSH-DBSCAN with geometrically scaling values of $\varepsilon$. This yields a sequence of clusterings which guarantees the approximation condition of Definition 2.6. To ensure that our output is a proper cluster hierarchy, we take successive *intersections* of these clusterings, a notion we define below.

**Definition 3.1** (Clustering intersection)**.** Let $\mathcal{C}$ and $\mathcal{C}'$ be two clusterings of a set $S$. We define $\mathcal{C} \cap \mathcal{C}'$ as the clustering of $S$ obtained by taking all nonempty intersections between a cluster in $\mathcal{C}$ and a cluster in $\mathcal{C}'$. The noise points of $\mathcal{C} \cap \mathcal{C}'$ are given by the union of the noise points of $\mathcal{C}$ and $\mathcal{C}'$.

The analysis of LSH-HDBSCAN (proof of Theorem 3.2) is deferred to Appendix A.2.

We note that finding $D_{\max}$ exactly takes quadratic time, but it suffices to find a value in the range $[D_{\max}, 2D_{\max}]$, which can be done in linear time.

---

**Algorithm 4** LSH-HDBSCAN

---

**Input:** Points $P$; density parameter $m$; approximation parameters $c, \gamma$; failure probability $\delta$
**Output:** A cluster hierarchy $\mathcal{T} = (\mathcal{C}_1, \dots \mathcal{C}_L)$ on $P$ with associated scales $\varepsilon_1, \dots, \varepsilon_L$.
1: Let $D_{\max}$ be the diameter of $P$ and let $\Delta$ be the aspect ratio
2: $L = 1 + \lceil \log_{1+\gamma}(\Delta) \rceil$
3: **for** $i$ in $[L]$ **do**
4: $\quad \varepsilon_i \leftarrow D_{\max} \cdot (1+\gamma)^{1-i}$
5: $\quad \mathcal{C}_i \leftarrow \mathcal{C}_{i-1} \cap \textsf{LSH-DBSCAN}(P, \varepsilon_i, m, c, \delta/L)$
6: **end for**
7: Return $\{(\mathcal{C}_i, \varepsilon_i) : i \in [L]\}$.

---

### 3.3 A Quadratic Lower Bound

The runtimes of LSH-DBSCAN and LSH-HDBSCAN approach $\tilde{O}(n^2)$ as the approximation factor $c$ tends to 1. We prove that this asymptotic behavior is unavoidable: assuming SETH (Impagliazzo & Paturi, 2001), near-quadratic time is required for sufficiently fine approximations to DBSCAN, even in Euclidean space. The proof is deferred to Appendix A.3.

**Theorem 3.3.** *Assuming* SETH, *for any $\alpha > 0$, there exists $\gamma > 0$ such that computing a $(1+\gamma)$-approximate* DBSCAN *clustering requires time $\Omega(n^{2-\alpha})$, even in the Euclidean space $\mathbb{R}^{O(\log n)}$. In particular, computing the exact* DBSCAN *clustering requires time $n^{2-o(1)}$.*

The lower bound immediately extends to HDBSCAN.

**Corollary 3.1.** *Assuming* SETH, *for any $\alpha > 0$, there exists $\gamma > 0$ such that computing a $(1+\gamma)$-approximate* HDBSCAN *clustering requires time $\Omega(n^{2-\alpha})$, even in the Euclidean space $\mathbb{R}^{O(\log n)}$. In particular, computing the exact* HDBSCAN *hierarchy requires time $n^{2-o(1)}$.*

## 4 EXPERIMENTAL EVALUATION

In this section, we evaluate the computational efficiency of LSH-DBSCAN on several benchmarks. Specifically, we analyze the tradeoff between computational speedup and cluster misalignment of LSH-DBSCAN relative to standard DBSCAN across different approximation factors $c$. Additional experiments and details are included in Appendix B.

**Setup.** We evaluate LSH-DBSCAN by measuring computational efficiency and cluster misalignment, which we specify below.

- **Computational efficiency:** We measure efficiency in terms of the total number of distance and hash computations performed by LSH-DBSCAN, compared against the number of distance computations in DBSCAN. This choice ensures that our experiments are independent of implementation and hardware details. It also aligns with our theoretical analysis in Section 3, which identifies distance and hash computations as the asymptotic bottleneck of LSH-DBSCAN. For completeness, raw runtime results are included in Appendix B.1.

- **Misalignment:** While LSH-DBSCAN is guaranteed to approximate the output of DBSCAN (Definition 2.3), we introduce a more interpretable notion of *cluster misalignment*. Informally, given two clusterings $\mathcal{C}$ and $\mathcal{C}'$, their misalignment is the fraction of misclassified points with respect to the optimal cluster matching. A precise definition of misalignment is given in Appendix B.2.

Our implementation uses the E2LSH hash family (Datar et al., 2004). All experiments were run on a machine with 12x AMD Ryzen 5 7640U core (4.9GHz), 16GB RAM, and kernel 6.16.7-arch1-1.

**Datasets.** We measure the computational efficiency and cluster misalignment of LSH-DBSCAN on four datasets: MNIST, Fashion-MNIST, ALOI, and GloVe.

**Parameter settings.** Aside from GloVe, we chose $\varepsilon$ and $m$ so that the output of exact DBSCAN is roughly consistent with ground truth clusters. Since GloVe has no underlying clustering, we choose the parameters to give a nontrivial clustering. In all tests, we set $\delta = 0.5$. Note that the correctness guarantees of Theorem 3.1 hold even when the hash repetition parameter $K$ in Algorithms 2 and 3 is set smaller than the theoretical value. In our experiments, scaling $K$ by factors of $0.8$ and $0.4$ in Algorithms 2 and 3, respectively, led to improved computation speedups.

Table 1: The DBSCAN parameters $\varepsilon$ and $m$ with the resulting number of clusters.

|  | $\varepsilon$ | $m$ | $|\mathcal{C}|$ |
|---|---|---|---|
| MNIST | 1000 | 100 | 2 |
| Fashion-MNIST | 800 | 10 | 35 |
| ALOI | 7500 | 3 | 29 |
| GloVe | 5 | 4 | 6 |

Table 2: Computation speedup and misalignment of LSH-DBSCAN for different approximation factors $c$. Each cell shows **comp. speedup / misalignment** relative to DBSCAN.

| $c$ | MNIST $N = 60000, d = 784$ | Fashion-MNIST $N = 60000, d = 784$ | ALOI $N = 24000, d = 27648$ | GloVe $N = 60000, d = 100$ |
|---|---|---|---|---|
| 2.0 | 6.24 / $\leq$0.001 | 3.23 / 0.008 | 0.78 / $\leq$0.001 | 1.10 / 0.007 |
| 3.0 | 19.19 / 0.002 | 15.72 / 0.079 | 3.75 / 0.006 | 8.28 / 0.044 |
| 4.0 | 20.58 / 0.002 | 34.69 / 0.114 | 5.63 / 0.016 | 20.91 / 0.061 |
| 5.0 | 35.65 / 0.007 | 52.45 / 0.114 | 11.03 / 0.015 | 35.97 / 0.063 |
| 6.0 | 13.63 / 0.003 | 75.53 / 0.128 | 17.63 / 0.034 | 45.38 / 0.055 |
| 7.0 | 20.33 / 0.005 | 82.25 / 0.115 | 9.09 / 0.53 | 61.20 / 0.069 |
| 8.0 | 27.59 / 0.009 | 103.55 / 0.123 | 11.55 / 0.031 | 78.36 / 0.082 |
| 9.0 | 36.51 / 0.013 | 122.18 / 0.132 | 15.10 / 0.041 | 67.34 / 0.059 |

**Discussion of results.** Across all benchmarks, both the computation speedup and misalignment tend to grow with the approximation factor $c$. The speedup is smallest on ALOI, consistent with its smaller dataset size (theoretically, our speedup ratio scales polynomially with $n$). On both ALOI and MNIST, the variation in speedup and misalignment across different $c$ is relatively mild, which may be related to MNIST having a stable cluster of 1's and ALOI having well-separated clusters. By contrast, GloVe and Fashion-MNIST exhibit more pronounced speedups and larger misalignments, perhaps owing to their weaker cluster structure under the given parameters.

**Visualization of the algorithm.** In Figure 1, we provide a visualization of how LSH-DBSCAN clusters a subset of ALOI data with different approximation factors $c$. This visualization aligns with the intuition that denser and more well-separated clusters tend to remain stable as $c$ increases.

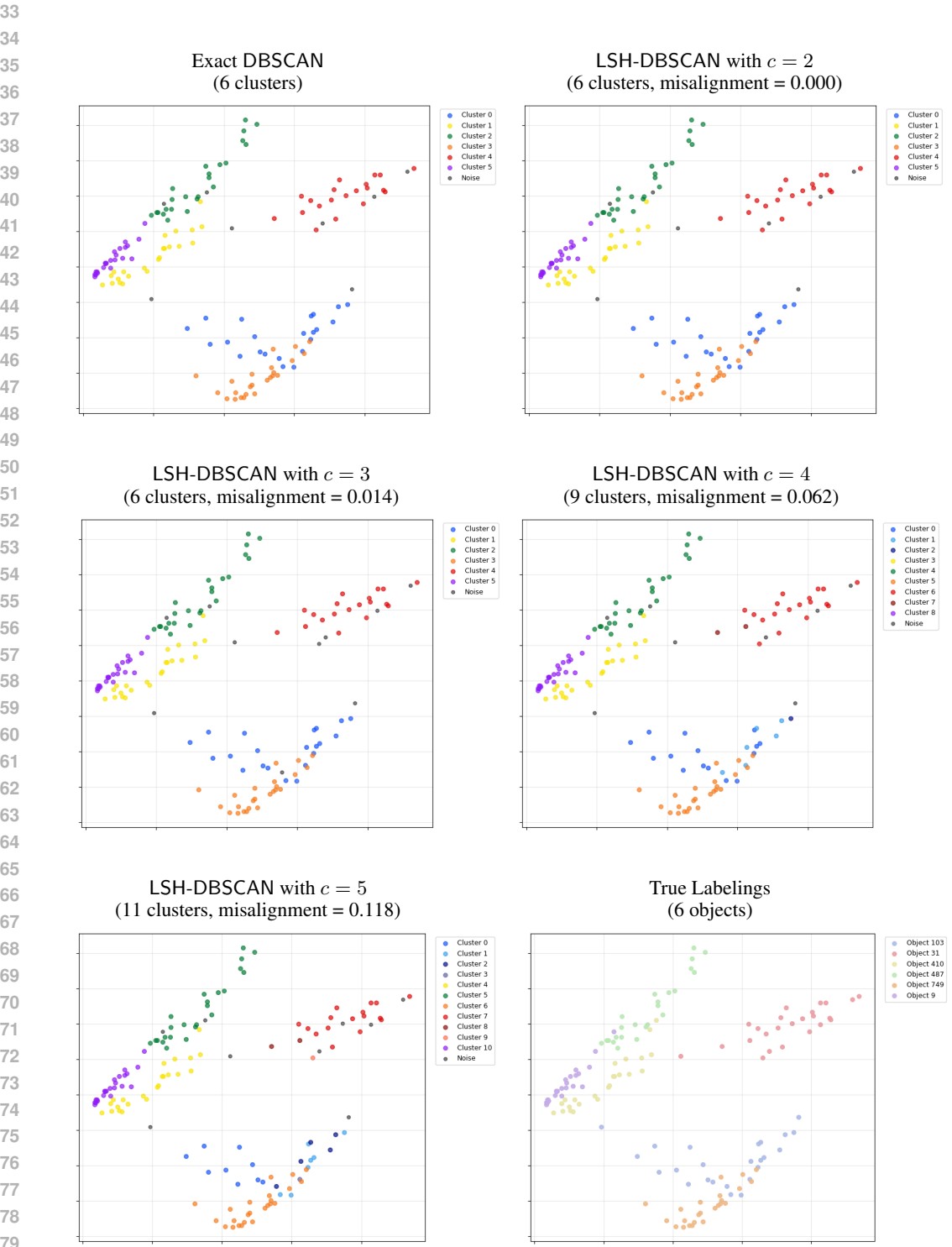

Figure 1: 2D PCA visualization of exact DBSCAN and LSH-DBSCAN on objects $\{103, 31, 410, 487, 749, 9\}$ of ALOI. We use parameters $\varepsilon = 7500$ and $m = 3$ across approximation factors $c \in \{2, 3, 4, 5\}$. Each plot includes the number of clusters and the misalignment score relative to exact DBSCAN. The final plot illustrates the ground truth labeling.

## 5 REPRODUCIBILITY STATEMENT

For our theoretical results in Section 3, we have included full proofs in Appendix A. To ensure reproducibility of our experimental results, we have attached a `.zip` file (anonymized) of all our experimental code as supplementary material. All random seeds have been fixed.

- For our runtime experiments, run

$$\texttt{python experiments/\{dataset\}\_runtime.py}$$

  for $\texttt{dataset} \in \{\texttt{mnist}, \texttt{fashion\_mnist}, \texttt{aloi}, \texttt{glove}\}$.

- For our ALOI visualization, run

$$\texttt{python experiments/aloi\_visualization.py}$$

- For our PCA failure experiment (Appendix B.3), run

$$\texttt{python experiments/pca\_failure.py}$$

.

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

## A  OMITTED PROOFS IN SECTION 3

In this appendix, we provide the full proofs of our algorithmic guarantees (Theorems 3.1 and 3.2) and our lower bound (Theorem 3.3).

### A.1  PROOF OF THEOREM 3.1

We first prove that Algorithm 2 satisfies the desired runtime and approximation guarantee. We note that a very similar subroutine was analyzed in Okkels et al. (2025).

**Lemma A.1** (Lemma 4.4 of Okkels et al. (2025)). *The expected running time of Algorithm 2 is*

$$O\left((T_{\mathrm{dist}} + T_{\mathrm{hash}}) \cdot n^{1+\rho} m^{1-\rho} \log(n) \log(n/\delta)\right).$$

*Proof.* We first bound the time spent computing hashes. For each $p \in P$ and $i \in [T]$, we compute the hash value $\boldsymbol{h}_i(p)$ twice: once when building the hash tables $B_{\boldsymbol{h}_1}, \ldots, B_{\boldsymbol{h}_T}$, and once when iterating over all buckets containing $p$. Since $\boldsymbol{h}_i$ is the concatenation of $K$ hash functions, the total time spent doing hash computations is

$$O(T_{\mathrm{hash}} \cdot nKT).$$

We now consider the time spent computing distances. Consider the loop iteration corresponding to a point $p \in P$. To determine whether $p$ should be added to $\widehat{\mathrm{Core}}$, the algorithm only needs to find up to $m$ distinct points at distance at most $c\varepsilon$ from $p$. So, the number of distance computations we perform is at most the number of *false positives*: points in $B_{\boldsymbol{h}_1}(p) \cup \cdots \cup B_{\boldsymbol{h}_T}(p)$ at distance $> \varepsilon$ from $p$.

Since $\mathcal{H}$ is $(\varepsilon/c, \varepsilon, p_1, p_2)$-sensitive, it follows that any $p' \in P$ with $\mathrm{dist}(p, p') \geq \varepsilon$ will satisfy

$$\Pr[\boldsymbol{h}_i(p) = \boldsymbol{h}_i(p') \text{ for some } i \in [T]] \leq T p_2^K = Tm/n$$

Therefore, the expected number of false positives for a given $p \in P$ is at most $O(Tm)$. We conclude that the time spent computing distances is at most

$$O(T_{\mathrm{dist}} \cdot Tnm).$$

Thus, the expected runtime of Algorithm 2 is at most:

$$O\left((T_{\mathrm{dist}} + T_{\mathrm{hash}}) \cdot TKnm\right) = O\left((T_{\mathrm{dist}} + T_{\mathrm{hash}}) \cdot n^{1+\rho} m^{1-\rho} \log(n) \log(n/\delta)\right),$$

where we used that $m \leq n$ and $K = O(\log n)$. □

**Lemma A.2** (Lemma 4.1 of Okkels et al. (2025)). *Let $\widehat{\mathrm{Core}}$ be the set of points returned by Algorithm 2. Let $\mathrm{Core}_{\varepsilon/c}$ and $\mathrm{Core}_\varepsilon$ denote the set of core points in the $(\varepsilon/c, m)$- and $(\varepsilon, m)$-DBSCAN clusterings of $P$, respectively. With probability at least $1 - \delta/2$, we have $\mathrm{Core}_{\varepsilon/c} \subseteq \widehat{\mathrm{Core}} \subseteq \mathrm{Core}_\varepsilon$.*

*Proof.* The second inclusion $\widehat{\mathrm{Core}} \subseteq \mathrm{Core}_\varepsilon$ is deterministically true, since the algorithm only includes a point $p$ in $\widehat{\mathrm{Core}}$ if it has found at least $m$ distinct points within distance $\varepsilon$ of $p$.

It remains to prove the first inclusion $\mathrm{Core}_{\varepsilon/c} \subseteq \widehat{\mathrm{Core}}$. Since $\mathcal{H}$ is $(\varepsilon/c, \varepsilon, p_1, p_2)$-sensitive, we have that for any pair of points $p, p'$ where $\mathrm{dist}(p, p') \leq \varepsilon/c$,

$$\Pr[\boldsymbol{h}_i(p) \neq \boldsymbol{h}_i(p') \text{ for all } i \in [T]] \leq (1 - p_1^K)^T \leq \exp\left(-T p_1^K\right) = \frac{\delta}{2nm}.$$

There are at most $n$ points in $\mathrm{Core}_{\varepsilon/c}$. For each such point, we include it in Core if we find at least $m$ of its neighbors. So, by a union bound over $nm$ pairs, we include all of $\mathrm{Core}_{\varepsilon/c}$ with probability at least $1 - \delta/2$. □

We now prove that Algorithm 3 satisfies the desired runtime and approximation guarantees.

**Lemma A.3.** *The expected runtime of Algorithm 3 is $O\left((T_{\mathrm{dist}} + T_{\mathrm{hash}}) \cdot n^{1+\rho} \log(n/\delta)\right)$.*

*Proof.* We first bound the time spend computing hashes. For each $p \in \widehat{\mathrm{Core}}$ and $i \in [T]$, we compute the hash value $\boldsymbol{h}_i(p)$ at most twice: once when building the hash tables $B_{\boldsymbol{h}_1}, \ldots, B_{\boldsymbol{h}_T}$, and once more if $p$ is ever dequeued from $Q$ (which can happen only once, since $p$ is immediately removed from all hash tables). Thus, letting $N := |\widehat{\mathrm{Core}}|$, the time spent computing hashes is at most

$$O(T_{\mathrm{hash}} \cdot NKT).$$

We now bound the time spent computing distances. Each point $p \in P$ is enqueued to $Q$ at most once, so the total number of distance computations $\mathsf{d}(u, v)$ where the distance is less than $\varepsilon$ is at most $N$. It remains to bound the number of *false positive* pairs $(u, v)$: those that are at distance $> \varepsilon$ but are hashed together in some table $B_{\boldsymbol{h}_i}$. Since $\mathcal{H}$ is $(\varepsilon/c, \varepsilon, p_1, p_2)$ sensitive, for any $\mathrm{dist}(u, v) \geq \varepsilon$, we have

$$\Pr[\boldsymbol{h}_i(u) = \boldsymbol{h}_i(v) \text{ for some } i \in [T]] \leq T p_2^K = T/N.$$

So, the expected number of false positive pairs is at most $O(TN)$. We conclude that the total amount of time spent computing distances is

$$O(T_{\mathrm{dist}} \cdot TN).$$

Lastly, we observe that given a point $p \in P$ and its hash values $\boldsymbol{h}_1(p), \ldots, \boldsymbol{h}_T(p)$, removing $p$ from all hash tables $B_{\boldsymbol{h}_1}, \ldots, B_{\boldsymbol{h}_T}$ can be supported in time $O(NT \log n)$ by storing each hash bucket as a binary search tree. We conclude that the expected runtime of Algorithm 3 is at most:

$$O\left((T_{\mathrm{dist}} + T_{\mathrm{hash}} + 1) \cdot NKT\right) = O\left((T_{\mathrm{dist}} + T_{\mathrm{hash}}) \cdot n^{1+\rho} \log(n) \log(n/\delta)\right).$$

Here, we use that $N \leq n$ and $K = O(\log n)$. $\qquad\square$

**Lemma A.4.** *Let $C_1, \ldots, C_k$ be the clusters produced by Algorithm 3 using $\widehat{\mathrm{Core}}$ as the set of core points. Define $G_{\varepsilon/c}$ and $G_\varepsilon$ as the graphs on $\widehat{\mathrm{Core}}$ where edges connect pairs of points at distance less than $\varepsilon/c$ and $\varepsilon$, respectively. Then, with probability at least $1 - \delta/2$,*

- *any two points in the same connected component of $G_{\varepsilon/c}$ are in the same cluster,*

- *and any two points in different connected components of $G_\varepsilon$ are in different clusters.*

*Proof.* The proof is analogous to that of Lemma A.2. We only add a point $u$ to a cluster $C_\ell$ if there is an existing point in $C_i$ at distance $\leq \varepsilon$ from $u$. Therefore, if two points are in the same cluster $C_\ell$, then they must be in the same connected component of $G_\varepsilon$, proving the second bullet point.

It remains to prove the first bullet point. Let $N := |\widehat{\mathrm{Core}}|$. Since $\mathcal{H}$ is $(\varepsilon/c, \varepsilon, p_1, p_2)$-sensitive, it follows that for any pair of points $u, v \in P$ where $\mathrm{dist}(u, v) \leq \varepsilon/c$,

$$\Pr[\boldsymbol{h}_i(u) \neq \boldsymbol{h}_i(v) \text{ for all } i \in [T]] \leq (1 - p_1^K)^T \leq \exp\left(-T p_1^K\right) = \frac{\delta}{2N}.$$

Now, take any spanning forest on $G_{\varepsilon/c}$, and observe that its connected components are identical to those of $G_{\varepsilon/c}$. By a union bound over all $\leq N$ edges $(u, v)$ in the spanning forest, the probability that all pairs $(u, v)$ share a hash bucket is at least $1 - \delta/2$. Conditioned on this event, each connected component of $G_{\varepsilon/c}$ will lie within a single cluster $C_\ell$, proving the first bullet point. $\qquad\square$

Combining the above lemmas, we obtain the proof of our main theorem.

**Theorem 3.1.** *Let $P$ be an $n$-point dataset that is LSH-friendly with quality $\rho$. For any parameters $m \leq n$, $\varepsilon > 0$, $c \geq 1$, and $\delta > 0$, there is an algorithm that runs in expected time*

$$O\left((T_{\mathrm{dist}} + T_{\mathrm{hash}}) \cdot n^{1+\rho} m^{1-\rho} \log(n) \log(n/\delta)\right)$$

*and outputs a $c$-approximate $(\varepsilon, m)$-DBSCAN clustering of $P$, with probability at least $1 - \delta$.*

*Proof of Theorem 3.1.* The runtime guarantee follows from Lemmas A.1 and A.3. The approximation guarantee follows from Lemmas A.2 and A.4. $\qquad\square$

A.2   PROOF OF THEOREM 3.2

We begin with the following lemma regarding clustering intersections.

**Lemma A.5.** *If a clustering is a refinement of both $\mathcal{C}$ and $\mathcal{C}'$, then it is a refinement of $\mathcal{C} \cap \mathcal{C}'$. If either $\mathcal{C}$ or $\mathcal{C}'$ is a refinement of a clustering, then so is $\mathcal{C} \cap \mathcal{C}'$.*

*Proof.* Suppose a clustering $\mathcal{A}$ is a refinement of $\mathcal{C}$ and $\mathcal{C}'$. Then, any two points in the same cluster of $\mathcal{A}$ must be in the same cluster of $\mathcal{C}$ and in the same cluster of $\mathcal{C}'$. Thus, they are in the same cluster of $\mathcal{C} \cap \mathcal{C}'$. Noise points of $\mathcal{C}$ or $\mathcal{C}'$ must be noise points in $\mathcal{A}$, so noise points of $\mathcal{C} \cap \mathcal{C}'$ must also be noise points in $\mathcal{A}$. So, $\mathcal{A}$ is a refinement of $\mathcal{C} \cap \mathcal{C}'$.

Now, suppose instead that $\mathcal{C}$ is a refinement of $\mathcal{A}$. Any two points in the same cluster of $\mathcal{C} \cap \mathcal{C}'$ must be in the same cluster of $\mathcal{C}$ and thus also in the same cluster of $\mathcal{A}$. Any noise point of $\mathcal{A}$ must be a noise point of $\mathcal{C}$ and thus also a noise point of $\mathcal{C} \cap \mathcal{C}'$. □

We are now ready to prove our main theorem on LSH-HDBSCAN.

**Theorem 3.2.** *Let $P$ be an $n$-point dataset that is LSH-friendly with quality $\rho$ and has aspect ratio $\Delta$. For any parameters $m \leq n$, $c \geq 1$, and $\delta, \gamma > 0$, there is an algorithm that runs in expected time*

$$O\left((T_{\text{dist}} + T_{\text{hash}}) \cdot n^{1+\rho} m^{1-\rho} \log(n) \log(n/\delta) \cdot \log_{1+\gamma} \Delta\right)$$

*and outputs a $c(1 + \gamma)$-approximate $m$-HDBSCAN hierarchy on $P$, with probability at least $1 - \delta$.*

*Proof.* The main loop of Algorithm 4 runs $O(\log_{1+\gamma}(\Delta))$ times, so the runtime bound follows directly from Theorem 3.1 and the fact that the intersection of two clusterings can be computed in $O(n)$ time. Moreover, the output is a cluster hierarchy since it is given by successive intersections of clusterings. All that remains to prove is the approximation guarantee.

By a union bound, with probability $1 - \delta$, all calls to LSH-DBSCAN return a $c$-approximate DBSCAN clustering; we condition on this event for the remainder of the proof.

Let $\varepsilon > 0$ be given, and for each $r \in (0, \infty)$, let $\mathcal{A}_r$ denote the exact $(r, m)$-DBSCAN clustering of $P$. We first consider two trivial cases for $\varepsilon$:

- If $\varepsilon < \varepsilon_L$, then we may set $\sigma(\varepsilon) = L$: both $\mathcal{A}_\varepsilon$ and $\mathcal{C}_L$ are equal to the empty clustering

- If $\varepsilon > \varepsilon_1$, then we may set $\sigma(\varepsilon) = 1$: both $\mathcal{A}_\varepsilon$ and $\mathcal{C}_1$ are equal to the clustering $\{P\}$ since $m \leq n$

Finally, suppose $\varepsilon \in [\varepsilon_L, \varepsilon_1]$. We define $\sigma(\varepsilon)$ to be the largest index $i \in [L]$ for which $\varepsilon_i \leq \varepsilon$. By construction, we have

$$\varepsilon/(1 + \gamma) \leq \varepsilon_i \leq \varepsilon.$$

We claim that $\mathcal{C}_i$ is a $c(1 + \gamma)$-approximate $(\varepsilon, m)$-DBSCAN clustering. To see this, let $\mathcal{B}_i$ denote the output of LSH-DBSCAN$(P, \varepsilon_i, m, c, \delta/L)$. For the upper refinement, we have

$$\mathcal{C}_i \preceq \mathcal{B}_i \preceq \mathcal{A}_{\varepsilon_i} \preceq \mathcal{A}_\varepsilon.$$

For the lower refinement, we use the fact that $\mathcal{A}_{\varepsilon_i/c} \preceq \mathcal{B}_j$ for all $j \leq i$, which implies

$$\mathcal{C}_i = (\mathcal{B}_1 \cap \cdots \cap \mathcal{B}_i) \succeq \mathcal{A}_{\varepsilon_i/c} \succeq \mathcal{A}_{\varepsilon/c(1+\gamma)}.$$

This completes the proof. □

A.3   PROOF OF THEOREM 3.3

Here, we prove our near-quadratic time conditional lower bound against DBSCAN by reducing from the bichromatic closest pair problem.

**Definition A.1** (Approximate Bichromatic Closest Pair). Let $(X, \mathsf{d})$ be a metric space and let $A, B \subset X$ be sets of size $n$. Let $\mathsf{d}_{\min}(A, B)$ denote the BCP *distance*, defined as

$$\mathsf{d}_{\min}(A, B) := \min_{(a^*, b^*) \in A \times B} \mathsf{d}(a^*, b^*).$$

For $\gamma \geq 0$, the $(1 + \gamma)$-approximate *bichromatic closest pair* problem (BCP) asks for a pair $(a, b) \in A \times B$ satisfying $\mathsf{d}(a, b) \leq (1 + \gamma) \cdot \mathsf{d}_{\min}(A, B)$.

Rubinstein (2018) proved a near-quadratic time lower bound against any algorithm that computes a $(1 + \gamma)$-approximation of the BCP distance.

**Lemma A.6** (Theorem 4.1 of Rubinstein (2018)). *Assuming* SETH, *for every $\alpha > 0$, there exists $\gamma > 0$ such that the following holds. Given two sets $A, B \subset \{0, 1\}^{O(\log n)}$, each of size $n$, computing a $(1 + \gamma)$-approximation to $\mathsf{d}_{\min}(A, B)$ under the Euclidean metric requires time $\Omega(n^{2-\alpha})$.*

**Lemma A.7.** *Let $\gamma \in (0, 1)$. Suppose there is an algorithm that, for any $n$-point set in a metric space $(X, \mathsf{d})$ with aspect ratio $\Delta$, computes a $(1 + \gamma)$-approximate $(\varepsilon, 2)$-DBSCAN clustering in time $T(n, \Delta)$. Then, there exists an algorithm which computes a $(1 + O(\gamma))$-approximation to the BCP distance in $(X, \mathsf{d})$ in time*

$$O\left((T(n, \Delta) + n) \cdot \frac{\log \Delta}{\gamma}\right).$$

*Proof.* Let $\mathcal{A}$ be an algorithm for computing a $(1 + \gamma)$-approximate $(2, \varepsilon)$-DBSCAN clustering. Given size-$n$ subsets $A, B \subset X$, we compute a $(1 + \gamma)$-approximation to the BCP distance $\mathsf{d}_{\min}(A, B)$ as follows.

1. Arbitrarily choose a pair $(a, b) \in A \times B$ and set $i = 0$.

2. Repeat the following process:

    (a) Compute $\varepsilon_i := \mathsf{d}(a, b)/(1 + \gamma)^i$.
    (b) Run $\mathcal{A}$ to obtain $\mathcal{C}^{(i)}$, a $(1 + \gamma)$-approximate $(\varepsilon_i, 2)$-DBSCAN clustering of $A \cup B$.
    (c) If no cluster in $\mathcal{C}^{(i)}$ contains both a point in $A$ and a point in $B$, return $\varepsilon_i \cdot (1 + \gamma)$.
    (d) Otherwise, increment $i \leftarrow i + 1$ and repeat.

Let $\eta$ denote the output of the above algorithm. We will first show that $\eta$ satisfies the desired approximation guarantee. Observe that if $\varepsilon_i \geq \mathsf{d}_{\min}(A, B) \cdot (1 + \gamma)$, then there is a pair $(a^*, b^*) \in A \times B$ at distance $\mathsf{d}(a^*, b^*) \leq \varepsilon_i/(1+\gamma)$. Therefore, any approximate $(2, \varepsilon_i)$-DBSCAN clustering of $A \cup B$ must place $a^*$ and $b^*$ in the same cluster. From this we conclude that $\eta < \mathsf{d}_{\min}(A, B) \cdot (1+\gamma)^2$.

Next, observe that if $\varepsilon_i < \mathsf{d}_{\min}$, then in any $(1+\gamma)$-approximate $(2, \varepsilon_i)$-DBSCAN clustering of $A \cup B$, no clusters contain both a point in $A$ and a point in $B$. From this we conclude that $\eta \geq \mathsf{d}_{\min}(A, B)$. Combining the two observations, we have

$$\eta/\mathsf{d}_{\min}(A, B) \in [1, (1 + \gamma)^2].$$

Therefore, $\eta$ is a $(1 + \gamma)^2 = (1 + O(\gamma))$-approximation of $\mathsf{d}_{\min}(A, B)$.

To bound the runtime of this algorithm, observe that each iteration of Step 2 takes $T(n, \Delta)$ time to produce the clustering $\mathcal{C}^{(i)}$ and $O(n)$ time to determine whether any cluster in $\mathcal{C}^{(i)}$ contains a pair in $A \times B$. The number of iterations of step 2 is at most $\log_{1+\gamma} \Delta = O\left(\frac{\log \Delta}{\gamma}\right)$. $\qquad\square$

**Theorem 3.3.** *Assuming* SETH, *for any $\alpha > 0$, there exists $\gamma > 0$ such that computing a $(1 + \gamma)$-approximate* DBSCAN *clustering requires time $\Omega(n^{2-\alpha})$, even in the Euclidean space $\mathbb{R}^{O(\log n)}$. In particular, computing the exact* DBSCAN *clustering requires time $n^{2-o(1)}$.*

*Proof.* Let $\alpha > 0$. Combining Lemmas A.6 and A.7, there is a constant $\gamma > 0$ such that the following holds: under SETH, any algorithm for solving $(1 + \gamma)$-approximate DBSCAN on inputs

in $\{0, 1\}^{O(\log n)}$ requires time

$$\Omega\left(\frac{n^{2-\alpha}}{\gamma \log \Delta}\right) = \Omega(n^{2-\alpha-o(1)}),$$

since $\log \Delta = O(\log \log n)$. The theorem follows by a parameter change in $\gamma$. $\qquad\square$

# B    ADDITIONAL EXPERIMENTAL DETAILS

## B.1    RUNTIME COMPARISON RESULTS

For completeness, we include raw runtime speedup results for our experiment in Section 4.

Table 3: Runtime speedup and cluster misalignment of LSH-DBSCAN for different approximation factors $c$. The first row contains runtimes of DBSCAN in seconds. All other cells show **time speedup / misalignment** relative to the exact clustering.

| $c$ | MNIST $N = 60000, d = 784$ | Fashion-MNIST $N = 60000, d = 784$ | ALOI $N = 24000, d = 27648$ | GloVe $N = 60000, d = 100$ |
|---|---|---|---|---|
| exact | 22402.55s | 13839.67s | 9092.66s | 11467.41s |
| 2.0 | 15.83 / ≤0.001 | 5.45 / 0.008 | 0.34 / ≤0.001 | 1.04 / 0.007 |
| 3.0 | 26.88 / 0.002 | 31.00 / 0.079 | 1.63 / 0.006 | 10.35 / 0.044 |
| 4.0 | 38.27 / 0.002 | 69.42 / 0.114 | 3.25 / 0.016 | 33.46 / 0.061 |
| 5.0 | 66.03 / 0.007 | 83.45 / 0.114 | 5.94 / 0.015 | 39.48 / 0.063 |
| 6.0 | 22.31 / 0.003 | 139.55 / 0.128 | 8.71 / 0.034 | 51.82 / 0.055 |
| 7.0 | 33.82 / 0.005 | 45.20 / 0.115 | 6.19 / 0.53 | 76.35 / 0.069 |
| 8.0 | 47.36 / 0.009 | 158.64 / 0.123 | 7.66 / 0.031 | 49.84 / 0.082 |
| 9.0 | 64.02 / 0.013 | 204.14 / 0.132 | 9.75 / 0.041 | 50.80 / 0.059 |

## B.2    FORMAL NOTION OF MISALIGNMENT

Here, we formalize our notion of misalignment between clusterings, which captures the fraction of misclassified points.

**Definition B.1.** Let $\mathcal{C} = \{C_1, \ldots, C_k\}$ be a clustering of a set $S$. For $p \in S$, define $\mathcal{C}(p) = i$ where $p \in C_i$, and $\mathcal{C}(p) = 0$ if no such $i$ exists.

**Definition B.2** (Misalignment). Given two clusterings $\mathcal{C} = \{C_1, \ldots, C_k\}$ and $\mathcal{C}' = \{C'_1, \ldots, C'_{k'}\}$ of a set $S$, the misalignment of $\mathcal{C}'$ and $\mathcal{C}$ is

$$\min_{\pi}\big|\{p \in S : \mathcal{C}(p) \neq \pi(\mathcal{C}'(p))\}\big|$$

where the minimum is over all permutations $\pi : [\max(k, k')] \to [\max(k, k')]$.

## B.3    FAILURE OF DIMENSION REDUCTION

While our paper focused on subquadratic-time algorithms for approximating DBSCAN in high dimensions, significantly faster algorithms are known for low-dimensional settings, even for exact DBSCAN (Gan & Tao, 2017; Wang et al., 2020). Therefore, it is common in practice to first apply dimension reduction (e.g., PCA, UMAP) and then run a fast low-dimensional DBSCAN. However, dimensionality reduction is known to distort neighborhood structures (Snoeck et al., 2025), suggesting that such methods may not always yield accurate clusterings.

We illustrate this failure of dimension reduction with a simple dataset where PCA performs poorly. The dataset (Figure 2) consists of $n = 1000$ images, each a $100 \times 100$ white grid containing a randomly shifted square. Half of the images contain a black $50 \times 50$ square, and the other half contain a gray $60 \times 60$ square. Because of their difference in size and brightness, running DBSCAN

in the original ambient space ($d = 10000$) easily separates the two clusters. By contrast, a two-dimensional PCA substantially mixes these two classes (Figure 3). In fact, we show that even a 12-dimensional PCA distorts the points enough to prevent DBSCAN from recovering the original clustering.

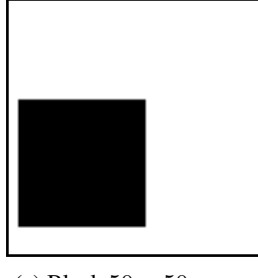

(a) Black $50 \times 50$ square

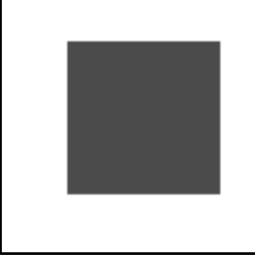

(b) Gray $60 \times 60$ square

Figure 2: **Synthetic squares dataset.** Consists of 1000 randomly shifted squares, 500 black and 500 gray.

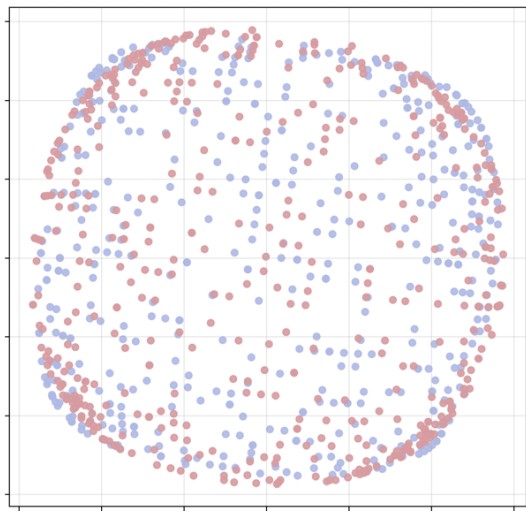

Figure 3: Two-dimensional PCA projection of the synthetic squares dataset. The two classes (black $50 \times 50$ vs. gray $60 \times 60$ squares) mix heavily, showing that PCA fails to separate them.

Table 4: Exact DBSCAN misalignment ($m = 3$) on the synthetic squares dataset. Misalignment is measured relative to the ground truth clustering.

| Dataset | $\varepsilon$ Range | Max Cluster Size | Misalignment |
|---|---|---|---|
| Squares (original, $m = 3$) | $[3300, 5500]$ | 500 | 0 |
| Squares (12D PCA, $m = 3$) | $[600, 607]$ | – | $\geq 0.486$ |
| | $\leq 600$ | 277 | $\geq 0.446$ |
| | $\geq 607$ | 754 | $\geq 0.254$ |

Table 4 summarizes the misalignment of exact DBSCAN (with $m = 3$) on both the original dataset and its 12-dimensional PCA projection. The misalignment bounds were computed as follows:

- **Original dataset.** For $\varepsilon \in \{3300, 5500\}$, running $(\varepsilon, m)$-DBSCAN recovers the true clustering exactly, which implies that the misalignment is 0 for all $\varepsilon \in \{3300, 5500\}$.
- **12D PCA projection.** For $\varepsilon = 600$ and $\varepsilon = 607$, running $(\varepsilon, m)$-DBSCAN produces clusterings with maximum cluster sizes 277 and 754, respectively. This implies that for

all $\varepsilon \leq 600$ and $\varepsilon \geq 607$, the maximum cluster size is either $\leq 277$ or $\geq 754$. Since the ground truth clustering consists of two clusters of size $500$, we have

$$\text{misalignment} \geq \max \left\{ 2 \cdot \left( 0.5 - \frac{\text{max cluster size}}{n} \right), \ \frac{\text{max cluster size}}{n} - 0.5 \right\},$$

which gives the indicated bounds. For $\varepsilon \in [600, 607]$, we iterated over all pairwise distances in this range and observed that the minimum misalignment is $0.486$.

One popular alternative to PCA is UMAP (McInnes et al., 2018), a dimension reduction technique that focuses on preserving local distances. However, few theoretical guarantees are known for UMAP. Moreover, its time complexity is bottlenecked by an approximate nearest neighbor computation at every dataset point, and thus it inherits the same theoretical limitations as LSH-DBSCAN.

