# OpenReview forum: "Provably Fast Density-Based Clustering in High Dimensions"
_ICLR.cc/2026/Conference — Submitted to ICLR 2026_

### Official Review · Reviewer_TiWK · 2025-10-28

**Soundness:** 2
**Presentation:** 1
**Contribution:** 2
**Rating:** 2
**Confidence:** 4

**Summary:**

This paper proposes LSH-DBSCAN and LSH-HDBSCAN which are approximate versions of DBSCAN and HDBSCAN that can achieve provably subquadratic running time on high dimensional datasets. The main idea from the authors is to identify the core points via the LSH buckets and then form the clusters by an LSH assisted BFS that expands only within the buckets. The authors are proving this using theorems 3.1 and 3.2 along with the experimental results that were run on datasets MNIST, Fashion MNIST, ALOI and Glove. The experimental results were compared with exact DBSCAN and they reported the computation-count speedups and misalignment metric values which measures the discrepancy from exact DBSCAN.

**Strengths:**

The dataset assumptions are removed compared to Okkels et al paper.

The main algorithm is well written by dividing it into two phases making it easy to understand.

The preliminaries and theorems were also well written and explained and also proved in the appendix.

The SETH based lower bounds clarified that substantially better than quadratic exact algorithms are unlikely.

**Weaknesses:**

The authors compared the new algorithm only to exact DBSCAN. They were not compared to other implementations like DBSCAN++, DBSCAN with dimensionally reduced techniques like PCA/ UMAP, or HNSW. There are many algorithms claiming improved speed and accuracy in high dimensions.

The authors haven’t explained why there was a sudden decrease in the speedup in ALOI dataset when c = 7.0. Also, it shows a 53% misalignment which is very large compared to the other c value misalignments.
The authors haven’t mentioned anything regarding the space complexity and practical memory measurements.

The authors also use delta = 0.5 which is a very high failure probability value. They haven’t really explained why.

The authors also haven’t properly defined the variables. For example, in Definition 2.5, L is not defined. It is only later known what exactly is L.

There is nothing on practical performance. Which is unacceptable because there are already efficient version of dbscan available. The claim is on a provable speedup in high dimensions. But the algorithm is an approximate. So the claim should also be on accuracy with appropriate and fair comparisons.

Due to these important issues we think that this work is not of high impact at this current stage and more work is needed to highlight its importance.

**Questions:**

Could you please explain why there was a sudden decrease in speedup and also an increase in misalignment in the ALOI dataset when c=7.0?

Could you also include the comparisons with DBSCAN++, DBSCAN with dimensionally reduced techniques like PCA/ UMAP, or HNSW etc?

Could you also report the memory usage and also talk about the space complexity?

Also, could you provide a short theory or citation for the claimed linear time 2 approximation of Dmax in general metric spaces?

Also, could you please explain what is the base case C0? If Dmax gets over estimated, how does this affect the approximation mapping from definition 2.6?

Could you please explain the reason for choosing the value of 0.5 for failure probability value?

---

> ### Author Response · Authors · 2025-11-14
>
> We thank the reviewer for their thoughtful comments and for raising several important points.
>
> **Practical performance and existing heuristics**
>
> The goal of our experimental section is to demonstrate that it is possible to have a fast practical implementation of DBSCAN while maintaining worst-case approximation guarantees. Many existing accelerations, including DBSCAN++ and HNSW-based variants are heuristic approaches that do not offer worst-case guarantees on the output clustering. While it is popular to use PCA and UMAP to reduce dimensionality, this would also lose theoretical guarantees of DBSCAN (see Appendix B). Developing a fast implementation of our algorithm and comparing it against existing DBSCAN heuristics is an interesting direction for future work, but it is a different question than the one addressed in this paper.
>
> **ALOI anomaly at c = 7.0**
>
> Thank you for pointing out the anomaly. This was a typo: the correct misalignment value is 0.053 (this will be fixed in the next revision). More broadly, while our theoretical speedup guarantee increases with c, this guarantee is only an upper bound. Certain inputs and parameter choices can yield substantially better runtimes in practice. Nonetheless, the overall trend in our experiments is that speedup indeed increases with c.
>
> **Failure probability**
>
> We acknowledge that a smaller failure probability than delta=0.5 could have been used. However, the worst-case runtime of our algorithm scales with log(1/delta), so decreasing delta would only change the speedup by a small constant factor.
>
> **Definition of L**
>
> The meaning of L in Definition 2.5 is the number of unique pairwise distances in the pointset.
>
> **Space complexity and memory usage**
>
> The space complexity of our algorithms is bounded above by the time complexity, which is subquadratic in the input size.
>
> **2-approximation of Dmax and the base case**
>
> One can obtain a 2-approximation of the maximum distance in linear time by a simple triangle inequality argument: take any point and compute the distance to the furthest point from it. Starting with a factor-2 overestimate of Dmax in Algorithm 4 (LSH-HDBSCAN) contributes only a constant overhead to the runtime.

---

> > ### Comment · Reviewer_TiWK · 2025-11-26
> >
> > Thank you for responding and providing explanations.

---

### Official Review · Reviewer_E86Z · 2025-10-29

**Soundness:** 3
**Presentation:** 2
**Contribution:** 2
**Rating:** 4
**Confidence:** 4

**Summary:**

The paper presents fast algorithms for estimating DBSCAN* and HDBSCAN using LSH calculations.

**Strengths:**

The paper presents an algorithm to provably approximate DBSCAN and HDBSCAN clusterings in high-dimensional spaces, moving beyond the $O(dn^2)$ runtime to an $O(dn^{1 + 1/c^2})$ runtime for a $c$-approximation. This is a nice result and the speedups are correspondingly interesting. The experimental analysis is appropriate and the results are laid out in an intuitive manner.

**Weaknesses:**

I think this paper's primary weakness is that it operates within a vacuum and does not appropriately refer to the related work. Specifically, several of the results stated in the paper feel fairly straightforward as corollaries of known work but the present paper seems to re-derive them and present them as novel contributions. I present the intuition behind these ideas below.

First, there is the SIGMOD paper showing that HDBSCAN can be solved efficiently in a parallel setting [1]. Although the current work does not assume a parallel environment, many of the ideas around accelerating the connected component computation using fast distance estimation seems to carry over. Second, there is the work analyzing the theoretical properties of DBSCAN* (DBSCAN without border points) in [2]. This is the algorithm being analyzed in the present work as well, it seems. [2] shows that DBSCAN* operates under an ultrametric and implies that the bottleneck to calculating DBSCAN* clusterings is the calculation of this ultrametric. Thus, the authors' present work can be interpreted as a fast approximation of these ultrametric quantities. This then naturally coincides with [3], which shows that this ultrametric also extends to the HDBSCAN algorithm (and many other density-based clustering algorithms). Put simply, there is a set of references which suggest that producing the DBSCAN* and HDBSCAN clusterings reduces to calculating an MST under a specific distance metric.

The authors similarly omit any reference to calculating MSTs efficiently. It is known that EMSTs require $O(n^2)$ time, but that this can be accelerated using LSH and similar distance-estimation methods such as hierarchically well-separated trees (HSTs). [4] is an archetypal paper in this context, but there are many similar ones such as [5]. Indeed, [5] produces similar reductions to bichromatic matching as in the submitted paper.

As a consequence, it is difficult to understand whether the derivations in the present paper are re-discoveries of various pieces of known literature or if they are doing something new. It seems that the reductions have been shown in previous works and that the pieces exist around the literature. If the authors can convince me that what they have done is significantly dissimilar to the literature on (a) theoretical analyses of DBSCAN* and HDBSCAN, (b) calculations of MSTs and (c) existing reductions between these ideas and fast LSH or HST distance calculations, then I would be happy to raise my score. As it stands, however, I am unconvinced of the present paper's novelty.


References:
[1]: Wang, Yiqiu, et al. "Fast parallel algorithms for euclidean minimum spanning tree and hierarchical spatial clustering." Proceedings of the 2021 international conference on management of data. 2021.

[2]: Beer, Anna, et al. "Connecting the Dots--Density-Connectivity Distance unifies DBSCAN, k-Center and Spectral Clustering." Proceedings of the 29th ACM SIGKDD conference on knowledge discovery and data mining. 2023.

[3]: Draganov, Andrew, et al. "I Want'Em All (At Once)--Ultrametric Cluster Hierarchies." arXiv preprint arXiv:2502.14018 (2025).

[4]: March, William B., Parikshit Ram, and Alexander G. Gray. "Fast euclidean minimum spanning tree: algorithm, analysis, and applications." Proceedings of the 16th ACM SIGKDD international conference on Knowledge discovery and data mining. 2010.

[5]: Indyk, Piotr. "Algorithms for dynamic geometric problems over data streams." Proceedings of the thirty-sixth annual ACM Symposium on Theory of Computing. 2004.

**Questions:**

Most of my questions are implied throughout the "weaknesses" section. A few specific ones would be:
- Is it fair to say that the authors are presenting a fast mechanism for calculating the dc-dist between points?
- How does this differ from existing LSH- and HST-based MST calculation techniques?

Side note: I disagree with the statement that the results in the paper easily extend to the standard DBSCAN definition which also includes border points. Introducing border points breaks the ultrametric properties of the algorithm, making it (a) not compatible with HDBSCAN and (b) require additional computational steps. Although I think that DBSCAN* is a superior algorithm to DBSCAN, the authors present their work as if it is about DBSCAN. I disagree with this framing.

---

> ### Author Response · Authors · 2025-11-14
>
> We thank the reviewer for the thoughtful comments and for pointing out several connections that we will definitely incorporate into the revised related work section.
>
> As the reviewer mentioned, one can indeed reduce DBSCAN* to computing an approximate MST – this was our initial approach. Building an approximate MST in turn reduces to approximate bichromatic closest pair, which can be solved using LSH. In fact, an interesting implication of our lower bound proof (Theorem 3.3) is that, up to logarithmic factors, DBSCAN* and bichromatic closest pair are equivalent.
>
> However, these reductions come with many hidden logarithmic factors that make this approach impractical. For example, the standard reductions from MST to chromatic closest pair and from chromatic to bichromatic closest pair each contribute a log(n)-factor overhead. Our algorithm for DBSCAN* differs from the MST approach by focusing only on a single density threshold. This allows us to get a practical algorithm that has provably subquadratic runtime. Our HDBSCAN algorithm enjoys a similar practicality benefit because it builds the hierarchy directly using LSH at all relevant scales, rather than fully constructing an approximate MST. We will emphasize all of these points in the next revision.
>
> So, to directly address the reviewer’s questions: (1) our algorithm does not compute the dc-distance between points, and (2) we do not compute the MST, as discussed above.
>
> Regarding the extension of DBSCAN* to DBSCAN, we believe our approach carries over directly. The core point identification step remains unchanged. The only modification is that the LSH-assisted BFS will traverse over all points rather than only core points (but only core points will be added to the BFS queue). The reviewer is correct that HDBSCAN is not compatible with DBSCAN when border points are included. We chose to omit the * in the paper for readability.

---

### Official Review · Reviewer_UjXy · 2025-10-31

**Soundness:** 2
**Presentation:** 3
**Contribution:** 1
**Rating:** 2
**Confidence:** 4

**Summary:**

The authors present faster versions of DBSCAN and HDBSCAN based on LSH.
While the basic versions are indeed slow, there is already a lot of research that accelerated them which the authors do not mention or compare to. Note that already ten years ago there were versions of DBSCAN with less than O(n^2), e.g., AnyDBC [0].
The experimental evaluation is very weak and the related work section has just a few lines that are not sufficient to cover the extensive related work in the field.
The novelty is limited as LSH and DBSCAN both have been around for a while and DBSCAN has already been accelerated with LSH in other works, e.g., [1,2]. While the work is closely related to [3], the novel methods are not compared to it .
Furthermore, as the proofs are one of the main selling criteria, they should be contained in the main paper, at least as a proof sketch.


[0] Mai, S. T., Assent, I., & Storgaard, M. (2016, August). AnyDBC: An efficient anytime density-based clustering algorithm for very large complex datasets. In Proceedings of the 22nd ACM SIGKDD international conference on knowledge discovery and data mining (pp. 1025-1034).

[1] Shiqiu, Y., & Qingsheng, Z. (2019, October). DBSCAN clustering algorithm based on locality sensitive hashing. In Journal of Physics: Conference Series (Vol. 1314, No. 1, p. 012177). IOP Publishing.

[2] Keramatian, A., Gulisano, V., Papatriantafilou, M., & Tsigas, P. (2022, August). IP. LSH. DBSCAN: Integrated Parallel Density-Based Clustering Through Locality-Sensitive Hashing. In European Conference on Parallel Processing (pp. 268-284). Cham: Springer International Publishing.

[3] Okkels, C. B., Aumüller, M., Thomsen, V. B., & Zimek, A. (2025). High-dimensional density-based clustering using locality-sensitive hashing. In EDBT 2025 (pp. 694-706).

**Strengths:**

S1) The paper is well-written and easy to follow

S2) The speedup compared to baseline DBSCAN works very well and the method relies on less assumptions on the data than [3]

S3) Theoretically sound and proofs are given

**Weaknesses:**

W1) The experimental evaluation is not sufficient.

a) Datasets like MNIST that consist of Gaussian clusters (with varying density) are not a good fit to density-based clustering. There exist a lot of benchmark datasets that actually contain density-based clusters, e.g. the DERIC benchmark (https://github.com/deric/clustering-benchmark), even if they are only low dimensional. For high-dimensional data, the COIL datasets work or also video data. Using only four datasets is not sufficient to show the quality and runtime in a statistically relevant way.

b) There is no comparison to competitors. Even if there are proofs in the paper, it is still important to see the behavior of your algorithms compared to related methods: Only then users can make an informed decision whether they want to use the accelerated version.

W2) Related work is not discussed sufficiently, see in the summary.

W3) Selection of parameters is not clear. How were the values for epsilon and minPts chosen? The authors state that they chose parameter values such that the results were "roughly consistent with ground truth clusters" - according to which measure? How close is roughly consistent? Were there several such settings?

**Questions:**

Q1) How did you chose the values for parameters? (see W3)

Q2) How does your results compare in practice with similar methods (see Summary for suggestions)? In which (benchmark) datasets does it actually lead to an advantage to not have strong assumptions regarding the dataset? Can you provide real world datasets where the assumptions of [3] do not hold but your algorithm succeeds?

Q3) How did you choose the subset of ALOI data in the visualisation part?

---

> ### Author Response · Authors · 2025-11-14
>
> We thank the reviewer for their detailed comments and for pointing out several relevant works.
>
> **Related work**
>
> We agree that our current related work section is too brief. We will expand it to include AnyDBC [0] and earlier LSH-based DBSCAN variants [1,2]. We emphasize, however, that these works do not provide worst-case subquadratic guarantees for approximate DBSCAN in high-dimensional Euclidean space. Our focus is specifically on obtaining such guarantees while still achieving practical performance.
>
> We also acknowledge the close connection to [3]. While our core point identification step parallels theirs, our main contribution is a subquadratic routine for the cluster formation step, which is the more involved part of the algorithm and the bottleneck in their work.
>
> **Comparisons to competitors**
>
> The reviewer raises a reasonable point about including more empirical baselines. Our experimental goal was to show that the theoretical algorithm can be made practical, rather than to compete with heuristic accelerations. Many existing methods, including [0,1,2], rely on dataset-specific assumptions or do not offer worst-case guarantees on the output clustering. A full empirical comparison would answer a different question and is an interesting direction for future work.
>
> **Choice of datasets**
>
> Regarding the choice of datasets, we selected standard benchmarks commonly used in prior work on density-based clustering in high dimensions. We are happy to add COIL as suggested. As for the concern about using only a small number of datasets, any selection of datasets inevitably introduces biases. This is precisely why we view the worst-case analysis as valuable: it provides a universal guarantee on runtime and approximation.
>
> **Parameter selection**
>
> The reviewer asks how we chose epsilon and MinPts. For each dataset, we selected values for which exact DBSCAN produced a meaningful nontrivial clustering that aligned with the dataset’s semantic classes. Extreme parameter settings lead to trivial clusterings, while intermediate choices give very similar results, which is why we chose not to include additional parameter choices.
>
> **Structural assumptions of [3]**
>
> The reviewer also asks for examples where methods with structural assumptions, such as those in [3], do not apply. A simple example is when clusters are densely packed (if R core points have at least S neighbors within distance epsilon, then the runtime of the algorithm in [3] is at least R*S, which can be quadratic in the input size). We will state this more clearly in the next revision.
>
> **ALOI visualization subset**
>
> For the ALOI visualizations, we chose six clusters that (1) were recovered by exact DBSCAN, and (2) had diverse representation in a 2D PCA. The goal was simply to illustrate how larger values of c affect cluster structure.

---

> > ### Comment · Reviewer_UjXy · 2025-11-21
> >
> > Thanks for the answer. While some things are more clear now, the main problems remain and I'll keep my score.
> >
> > **Choice of datasets**
> >
> > Regarding your answer "As for the concern about using only a small number of datasets, any selection of datasets inevitably introduces biases." - That is exactly why one should show more datasets. One introduces even more bias to the experimental evaluation by choosing just a small number of datasets (especially if it is not known how/why they were chosen).
> >
> > Regarding using "standard benchmarks commonly used in prior work on density-based clustering in high dimensions": Can you cite these prior works, please?
> >
> >
> > **Parameter selection**
> >
> > "DBSCAN produced a meaningful nontrivial clustering that aligned with the dataset’s semantic classes" - Your choices need to be reproducible: How was this decided? Did you measure ARI or NMI between the clustering and ground truth? Or did someone just visually check if it looks good? What is meaningful? How many parameter settings (and which values/ranges) were tested for that? Did you follow any heuristics?

---

### Official Review · Reviewer_kogf · 2025-10-31

**Soundness:** 2
**Presentation:** 3
**Contribution:** 2
**Rating:** 2
**Confidence:** 3

**Summary:**

DBSCAN is a highly successful and well-known density-based clustering algorithm. However, due to its inherent computational complexity bottleneck, DBSCAN becomes impractical for large-scale datasets, particularly when the data reside in high-dimensional spaces. To address this limitation, the authors build upon the work of Okkels et al. (2025) and propose a c-approximation variant of DBSCAN, termed LSH-DBSCAN, which leverages the technique of locality-sensitive hashing (LSH) to achieve improved efficiency while maintaining approximate consistency with the results of the traditional DBSCAN in high-dimensional settings. Furthermore, the authors extend this approach to the hierarchical version and introduce LSH-HDBSCAN, which aims to provide an approximate yet computationally efficient counterpart to HDBSCAN with comparable theoretical guarantees.

**Strengths:**

**LSH-DBSCAN** and **LSH-HDBSCAN** are both highly intuitive and easy to understand, demonstrating strong reproducibility. Moreover, the authors provide solid theoretical backing for both methods, ensuring a certain level of cluster quality within a bounded computational complexity.

**Weaknesses:**

1. **Lack of methodological innovation**
   The DBSCAN algorithm consists of two main components: *core point identification* and *cluster formation*. The computational complexity challenge primarily arises from the *core point identification* step. However, the authors closely followed the method proposed by Okkels et al. (2025) for this step, showing limited originality in their methodological contribution.

2. **Problems in the experimental design**
   a) The experiments lack comparison with necessary baseline competitors. Many prior works have addressed improving the computational efficiency of DBSCAN, such as **DBSCAN++ [1]**, **sngDBSCAN [2]**, and more recently **sDBSCAN [3]** which specifically targets high-dimensional data. However, the authors only use the traditional DBSCAN as the baseline, failing to demonstrate the superiority of their proposed approach over existing methods.

   b) The two evaluation metrics used in the experiments — *speed up* and *misalignment* — are not only difficult to interpret but also unintuitive. For time efficiency, it would be more straightforward to directly report the total runtime from start to finish. For cluster quality assessment, the authors should consider using common metrics from prior works, such as **Adjusted Rand Index (ARI)**, **Adjusted Mutual Information (AMI)**, and **normalized mutual information (NMI)**.

   c) Although **LSH-HDBSCAN** is mentioned as one of the main contributions, there is no corresponding evaluation of it in the experimental section.

   d) More large-scale high-dimensional datasets are also preferred to be shown in the experiment section, such as Pamap2 and Mnist8m.



3. **Overclaiming issues**
   There are several instances of overclaiming throughout the paper. For example, in line 66, the authors state that their method “does not require any assumption on the dataset.” However, in Theorems 3.1 and 3.2, they explicitly assume that the dataset *P* is *LSH-friendly with quality $\rho$*. According to Definition 2.8, being LSH-friendly with quality $\rho$ requires satisfying $\rho(c) = \log(1/p_1) / \log(1/p_2)$.

   Second, in the abstract, the authors claim that their method provides “the *first provably* subquadratic runtime for approximate DBSCAN on arbitrary high-dimensional datasets.” However, Theorems 3.1 and 3.2 only guarantee *approximate results with high probability*, and similar-level theoretical guarantees have already been provided in previous works such as **sDBSCAN [3]**.

**Reference**

[1] Jang, J., & Jiang, H. (2019, May). DBSCAN++: Towards fast and scalable density clustering. In International conference on machine learning (pp. 3019-3029). PMLR.

[2] Jiang, H., Jang, J., & Lacki, J. (2020). Faster DBSCAN via subsampled similarity queries. Advances in Neural Information Processing Systems, 33, 22407-22419.

[3] Xu, H., & Pham, N. (2024). Scalable DBSCAN with random projections. Advances in Neural Information Processing Systems, 37, 27978-28008.

**Questions:**

According to Theorem 3.2, the time complexity of LSH-DBSCAN is $O(d \cdot n^{1 + 1/(2c^2 - 1) + o(1)})$, where a larger value of c should correspond to lower computational cost. However, although I do not fully understand how the *computational speedup* is calculated, the results in Table 2 for the ALOI dataset appear anomalous. Specifically, for the ALOI dataset, when \(c = 7.0\), the speedup is unexpectedly lower than that for both \(c = 6.0\) and \(c = 8.0\), while the misalignment is unusually high, reaching 0.53. Could you please explain why this is happened?

---

> ### Author Response · Authors · 2025-11-14
>
> We thank the reviewer for their detailed comments.
>
> **Theoretical results**
>
> We would like to address the reviewer’s concerns about the requirement that the dataset is LSH-friendly with quality rho<1. We clarify that the setting of our paper is high-dimensional Euclidean space, where this property holds for all datasets (see Fact 2.1). Thus, LSH-friendliness is not an assumption on the dataset, but rather the metric space. For the sake of completeness, we chose to write Theorems 3.1 and 3.2 for general LSH-friendly metrics.
>
> Regarding the reviewer’s comment that our approximation guarantee holds with *high probability*, we emphasize that this is a common feature of randomized algorithms. While previous works such as [1,2,3] give subquadratic algorithms for DBSCAN, they do not have similar-level theoretical guarantees. In particular, they provide guarantees only under strong assumptions on the dataset.
>
> We acknowledge that the core point identification step of our algorithm parallels that of Okkels et al. (2025). However, we provide a subquadratic routine for the cluster formation step, which is the more intricate component and constitutes the main bottleneck in their approach.
>
> **Experimental design**
>
> We chose not to include [1,2,3] as baselines for our experiments as they do not have theoretical guarantees on the output clustering. The goal of our experimental section is to demonstrate that it is possible to have a fast practical implementation of DBSCAN while maintaining worst-case approximation guarantees. Developing such an implementation that competes with existing heuristics is an interesting direction for future work.
>
> We acknowledge that our experimental results are limited to DBSCAN. Since our HDBSCAN algorithm works by making a small number of calls to DBSCAN, we chose not to include empirical HDBSCAN results.
>
> Regarding the suggestion to include larger-scale datasets such as Pamap2 and Mnist8m, the limiting factor was the cost of running exact DBSCAN. We used the largest instances for which exact DBSCAN completed in a reasonable amount of time. As for the concern about using only a small number of datasets, any selection of datasets inevitably introduces biases. This is precisely why we view the worst-case analysis as valuable: it provides a universal guarantee on runtime and approximation.
>
> **Evaluation metrics**
>
> Our metric for cluster misalignment is simply the fraction of misclassified points. When this quantity is small, it immediately implies that the ARI, AMI, and NMI are also small. Our notion of speedup is the improvement in the number of point comparisons. This metric is hardware-independent and reflects the efficiency of our algorithm rather than our specific implementation. Note that we do report results on total runtime from start to finish in Appendix B.
>
> **ALOI anomaly at c = 7.0**
>
> Thank you for pointing out the anomaly. This was a typo: the correct misalignment value is 0.053 (this will be fixed in the next revision). More broadly, while our theoretical speedup guarantee increases with c, this guarantee is only an upper bound. Certain inputs and parameter choices can yield substantially better runtimes in practice. Nonetheless, the overall trend in our experiments is that speedup indeed increases with c.

---

> > ### Comment · Reviewer_kogf · 2025-11-28
> >
> > We appreciated the details responding, however there are still few weakness was unaddressed:
> >
> > **Experiment metrics:**
> >
> > Your statement that “this immediately implies that the ARI, AMI, and NMI are also small” does not provide a convincing justification for why you do not directly use these metrics, and instead introduce Misalignment, a metric defined by the yourselves. In my view, including these common metrics would make it much easier to directly compare your results with prior work, as many related studies report these metrics as well.
> >
> > I know the table in Appendix B provides the total runtime, but the table is still quite confusing. Only DBSCAN’s runtime is shown in seconds, while the meaning of the “speed-up” values is ambiguous—whether they represent how many times faster the method is or a difference in seconds. Presenting the runtime in seconds for every methods would help to avoid the confusion in my perspective.
> >
> > **Experimental design**
> >
> > I do not find it convincing that you exclude other baselines (e.g., [1, 2, 3]) simply because they lack theoretical guarantees on clustering quality. In fact, to the best of my knowledge, sDBSCAN [3] does offer theoretical guarantees on clustering quality (similar to what you did), although it indeed assumes certain conditions on the dataset.
> >
> > I believe the current experimental setup does not clearly demonstrate the advantages of your method compared to existing approaches. As a potential user, it is unclear why I should choose your algorithm over others (e.g. sDBSCAN [3]). Since you claim that a key benefit of your method is that it does not require any assumptions on the dataset, it would be helpful to provide empirical results showing scenarios where your method performs well but existing methods fail under the same data.
> >
> > As these concerns remain unaddressed, I would like to keep the current score.

---

### Official Review · Reviewer_6f7K · 2025-10-31

**Soundness:** 2
**Presentation:** 2
**Contribution:** 1
**Rating:** 4
**Confidence:** 4

**Summary:**

The authors present a fast algorithm for the DBSCAN problem based on Locality-Sensitive Hashing (LSH). They provide a provably sub-quadratic runtime for computing an approximate DBSCAN clustering in high-dimensional settings. To further extend their approach, they also propose an efficient approximation of the popular hierarchical variant, HDBSCAN. The paper provides both theoretical guarantees and empirical evaluations on benchmark datasets, showing computational speedups while maintaining low clustering error.

**Strengths:**

This paper provides the provably subquadratic-time algorithm for approximate DBSCAN and HDBSCAN in arbitrary high-dimensional spaces. The DBSCAN and HDBSCAN algorithms are easy to understand and implement. Experiments on 4 real-world datasets show that the proposed methods achieve speedups with minimal loss in clustering accuracy.

**Weaknesses:**

1. The authors state in Lines 64–66 that their algorithm does not rely on any dataset-specific assumptions. However, the time complexity in Theorems 3.1 and 3.2 depend on the assumptions that both the MinPts parameter m and the aspect ratio Δ are constant. The aspect ratio is typically assumed to be polynomially bounded in the input size n [1], and in the worst case, it can be exponentially large, up to 2^{n^{o(1)}} [2].

2. Okkels et al. (2025) have already applied LSH techniques to accelerate DBSCAN. The authors adopt a similar approach and extends it to the hierarchical setting of HDBSCAN, which represents only an incremental improvement.

3. To address the challenges of high-dimensional or general metric space, Mo et al. (2024) [3] recently proposed both exact and c-approximate DBSCAN algorithms with provable linear-time guarantees under low intrinsic dimensionality assumptions. This prior work is neither cited nor discussed in the current paper, and it is also missing from the experimental evaluation.

[1] Bhaskara, A., Vadgama, S., and Xu, H. Greedy sampling for approximate clustering in the presence of outliers. In Proceedings of the 33rd International Conference on Neural Information Processing Systems, pp. 11146–11155, 2019.

[2] Cohen-Addad, V. Approximation schemes for capacitated clustering in doubling metrics. In Proceedings of the 31st Annual Symposium on Discrete Algorithms, pp. 2241–2259, 2020.

[3] Mo G, Song S, Ding H. Towards metric DBSCAN: exact, approximate, and streaming algorithms[J]. Proceedings of the ACM on Management of Data, 2024, 2(3): 1-25.

**Questions:**

1. Given that Δ can be as large as 2^{n^{o(1)}} in the worst case, how does the algorithm's runtime scale in such situations? Is the claimed sub-quadratic complexity still valid?

2. Could the authors provide empirical evidence that Δ remains small in the datasets used for evaluation? Otherwise, how can we be confident that the reported runtime reflects the theoretical guarantees?

3. The authors should provide comparative experiments with standard HDBSCAN.

4. The authors should include Mo et al. (2024) as a baseline to provide a more comprehensive evaluation.

---

> ### Author Response · Authors · 2025-11-14
>
> We thank the reviewer for their detailed comments.
>
> **Dataset assumptions**
>
> We would first like to address the reviewer’s comment regarding dataset assumptions. We don’t assume that the aspect ratio ∆ is constant. We clarify that MinPts is an input parameter of the algorithm, not an assumption on the data. We elaborate on both of these points below.
>
> The runtime of our DBSCAN algorithm *does not* depend on the aspect ratio ∆. This is why we did not present the empirical magnitude of ∆ in our experiments. The runtime of our HDBSCAN algorithm does depend on ∆. We emphasize that for a subquadratic runtime we only require that log(∆) is sublinear in the input size. Even ∆ as large as 2^{n^{o(1)}} satisfies this requirement.
>
> We acknowledge that our algorithm makes an assumption on the size of the MinPts parameter. For a subquadratic runtime, it suffices that MinPts = n^{o(1)} (in the literature [de Berg et al. (2017), Gan & Tao (2017)], it is standard to make the stronger assumption that MinPts is constant). When MinPts is large (e.g., linear in n), there is a simple quadratic lower bound on approximate DBSCAN. We will make sure to include these points in the next revision of our paper.
>
> **Other Comments**
>
> We thank the reviewer for mentioning Mo et al. (2024); we will make sure to include this reference in our Related Work section. We would like to emphasize that the relevant algorithms in Mo et al. have runtime that is *exponential* in the intrinsic dimensionality of the input pointset. Our paper focuses on the regime of high-dimensional Euclidean space, where the intrinsic dimensionality may be prohibitively large for such algorithms, so it is not applicable to use their algorithms as an empirical baseline for ours.
>
> We acknowledge that our experimental results are limited to DBSCAN. The purpose of our experimental section was to demonstrate that the theoretical algorithms in this paper can be made practical. Since our HDBSCAN algorithm works by making a small number of calls to DBSCAN, we did not see the need to include empirical HDBSCAN results.
>
> We acknowledge that the core point identification step of our algorithm parallels that of Okkels et al. (2025). While their primary contribution is a fast LSH-based *implementation* of approximate DBSCAN, our work provides a provably subquadratic algorithm for approximate DBSCAN, covering both core point identification and cluster formation. A secondary contribution of our work is empirical evidence that this theoretically grounded approach can be made practical.

---

> ### Comment · Reviewer_6f7K · 2025-11-22
> **Response to the Authors**
>
> **Follow-up on aspect ratio dependence**
>
> Thanks for the clarification about dataset assumptions. However, your sub-quadratic bound still requires $\log\Delta = n^{o(1)})$, and $\Delta$ can be arbitrarily large in the worst case as pointed out in the literature [1]-[2]. In that regime the runtime can again become very large. Recent work (e.g., Draganov et al. 2024 [3]) uses a preprocessing step to “compress’’ $\Delta$ to be polynomial in $n$ before running the main clustering algorithm. Can a similar compression or rounding scheme be integrated into your DBSCAN framework? If not, I would encourage the authors to state the guarantee more explicitly as "near-linear in the data size under a bounded (or compressed) aspect ratio", which would somewhat weaken the claimed contribution.
>
> **Other remaining concerns**
>
> - MinPts assumption: For sub-quadratic time you assume ($\text{MinPts} = n^{o(1)}$). This feels stronger than the standard "constant MinPts" assumption. Please clarify how realistic this is in practice and whether your guarantees extend when MinPts is larger. A detailed discussion on the connection between runtime complexity and MinPts ranges would make the theoretical contributions clearer.
>
> - Experimental scope. The experiments only evaluate approximate DBSCAN, without any empirical results for the proposed HDBSCAN algorithm or comparisons to existing HDBSCAN implementations. Since HDBSCAN also provides algorithmic contribution, I still believe at least an empirical comparison is important.
>
> **Relation to prior work.**
>
> For Mo et al. (2024), even if their algorithms are exponential in intrinsic dimension, it would help to more precisely compare guarantees and delineate the regimes where each method is preferable, rather than just stating that they are “not applicable’’ in high-dimensional regimes. In particular, what are the advantages of the proposed algorithms in low-dimensional settings?
>
> For Okkels et al. (2025), your core-point identification step is quite close to their LSH-based approach. A sharper technical comparison (what is reused vs. what is new) would make the contribution clearer.
>
> Overall, my key concerns are not fully addressed by the authors’ response. The strength and novelty of the paper are still not clear enough for me at this stage.
>
> [1] Nguyen H L, Nguyen T, Jones M. Fair range k-center[J]. arXiv preprint arXiv:2207.11337, 2022.
>
> [2] Bhattacharjee R, Moshkovitz M. No-substitution k-means clustering with adversarial order[C]//Algorithmic Learning Theory. PMLR, 2021: 345-366.
>
> [3] Draganov A, Saulpic D, Schwiegelshohn C. Settling time vs. accuracy tradeoffs for clustering big data[J]. Proceedings of the ACM on Management of Data, 2024, 2(3): 1-25.

---

### Meta-Review · Area_Chair_Yswh · 2026-01-05

**Summary:**

This paper studied the well-known density clustering model, DBSCAN, in high dimensional space. Relying on the LSH technique, the authors introduce LSH-DBSCAN and LSH-HDBSCAN. They also prove that their algorithms satisfy the formal approximation guarantee and have a subquadratic runtime.

There are five reviewers commented on this paper. Roughly speaking, the reviewers raised  two important issues: 1) the novelty, since the proposed method heavily rely on LSH and prior work Okkels et al. (2025); 2) the insufficient experimental part, lack a number of baselines as pointed by Reviewer 6f7K, Reviewer kogf, Reviewer TiWK.

In addition, I agree with Reviewer kogf on the overclaiming issues.

**Reviewer Concerns:**

The authors provide some explanation regarding the raised questions, but they did not full addressed them especially in terms of the novelty issue and the insufficient experiment issue.

**Reviewer Scores:**

The reviewers give 4, 2, 2, 4, 2. Except for Reviewer E86Z, all the other 4 reviewers participated the discussion during rebuttal, but I don't think they are willing to change their scores.

---

### Decision · Program_Chairs · 2026-01-26

Reject